# Effects of taping techniques on arch deformation in adults with pes planus: A meta-analysis

**Meihua Tang**[1], **Lin Wang**[1], **Yanwei You**[2], **Jiajia Li**[1], **Xiaoyue Hu**[1]*

**1** School of Kinesiology, Shanghai University of Sport, Shanghai, China, **2** Division of Sport Science & Physical Education, Tsinghua University, Beijing, China

☯ These authors contributed equally to this work.
* moonhxyy@163.com

**Data Availability Statement:** If the data are all contained within the manuscript and/or Supporting Information files, enter the following: All relevant data are within the manuscript and its Supporting Information files.

## Abstract

### Objective

To investigate effects of taping techniques on arch deformation in adults with pes planus.

### Methods

The following databases were searched up to March 2020, including Web of Science, Pubmed, EBSCO, CNKI and Cochrane Library. Heterogeneity and publication bias were assessed by $I^2$ index and funnel plots, respectively. In addition, Cochrane scale was used to evaluate the quality of research.

### Results

Navicular height for three antipronation taping techniques significantly increased immediately post tape compared with baseline (mean difference = 4.86 mm, 95% CI = 2.86–6.87 mm, Z = 4.75, p < 0.001). The highest increase was observed in Augmented low-Dye (ALD). Modified low-Dye (MLD) was second only to ALD (p<0.001). Navicular height after walking for 10 min was much higher than baseline (p<0.001), with MLD decreased smaller than ALD.

### Conclusions

ALD was the most effective taping technique for controlling foot arch collapse immediately post tape compared with baseline, followed by MLD. By contrast, MLD could possibly performed better than ALD in maintaining immediate navicular height after walking for 10 min. Low-Dye could make resting calcaneal stance position closer to neutral position. Although positive effects of Navicular sling, low-Dye and Double X taping interventions were observed, they could not maintain this immediate navicular height effect after a period of higher intensity weight-bearing exercise.

**Funding:** We declare that the authors received no specific funding for this work.

**Competing interests:** The authors have declared that no competing interests exist.

## Introduction

Pes planus is a foot arch deformity, is also known as flexible flat foot or planovalgus and is caused by talonavicular ligament laxity or foot arch intrinsic muscle weakness; it is character-ised by the navicular bone shifting inwards and downwards from the subtalar joint [1,2]. Foot arch plays an important role in cushioning ground impact and stabilising the body when standing and walking [3]. Adults with pes planus lack an elastic foot arch to attenuate the impact force [4]. This condition results in pain and impaired lower limb function, such as plantar fasciitis, plantar heel pain, posterior tibial stress syndrome and femoral patellar pain syndrome [5–8]. A previous study found that perimenopausal women with pes planus per-formed impaired postural balance compared with their counterparts with normal feet [9]. Pes planus is more common in adolescent males than females, and associated with higher BMI index, this condition even continues to adulthood without timely and effective intervention [10,11].

Currently, exercise interventions for pes planus focus on neuromuscular training, including arch doming, anterior and posterior tibial motor control training and towel curl exercises [12,13]. Ergonomic devices, such as kinesio taping, white athletic tape, arch support and motion control footwear are also used to treat pes planus [14–16]. Neuromuscular training has a long intervention period and takes effect slowly [12,17], orthoses are another common treat-ment for pes planus, such as arch supports and taping techniques [16,18,19]. However, studies showed that arch supports were not efficient in correcting the talus deviation [12,20]. Holmes et al. found that Modified low-Dye (MLD) could make the subtalar joints closer to the neutral position [21]. A meta-analysis was also conducted to determine the effects of taping, orthotics and motion control shoes on calcaneal eversion during gait in subjects with musculoskeletal conditions potentially related to excessive foot pronation [16]. The results showed that taping technique was the most effective methods to control calcaneal eversion. Therefore, athletic tap-ing, which is a non-invasive, simple and effective pes planus intervention method, has gradu-ally attracted the attention of researchers.

Among various taping techniques, white non-elastic adhesive tape material is mostly used for adults with pes planus. The main taping techniques include low-Dye, augmented low-Dye (ALD), MLD, Fan-arch support, Double X and Navicular sling [18,19,21–24]. The tape width is from 2.54 to 7.62 cm, and the materials are mostly non-elastic. The main difference is taping techniques. The original low-Dye technique starts from the fifth metatarsophalangeal joint, passes through the heel to the first metatarsal to establish an anchor band and then sticks some horizontal tapes from the lateral anchor to the medial [22]. ALD is based on low-Dye and adds 3 tapes to build the medial longitudinal arch and 2 calcaneal stabilisation belts, which is better in foot arch lifting [25,26]. Different from ALD taping method, MLD taping technique needs to place the foot to an appropriate position pre-tape [21]. Before taping, the forefoot undergoes manual eversion. The physical therapist flexes the first metatarsophalangeal joint, and the pull-ing force of the sticking on the lateral calcaneus provides the reaction force. This force is applied to keep the subtalar joint in a neutral position. In addition, the anchor of the Fan-arch support taping technique is pulled from the fifth metatarsal to the first metatarsal area, and several 6-shaped tapes are pulled around the calcaneus in the middle of the forefoot of the anchor belt [18]. Double X taping technique combines the improved Fan-arch support (the first strip is applied medial to the level of the first metatarsal, wrapped around the heel and attached on the plantar surface of the first metatarsal [24]. The second strip is applied lateral to the level of the fifth metatarsal, wrapped around the heel and attached on the plantar surface of the fifth metatarsal) and low-Dye taping techniques. Only the Navicular sling technique uses elastic tape to pull a tape around the sole of the race plantar from the dorsum of the midfoot to

the sole of the lateral plantar, and it bypasses the ankle joint and finishes above the lateral ankle [19]. Although white taping methods have various types, a consensus lacks on a taping technique that can maintain the immediate effect during long-term exercise to interfere with pes planus. Therefore, the effect of taping techniques on the arch deformation among individuals with pes planus should be investigated.

This systemic review with meta-analysis aimed to quantitatively evaluate the efficacy of different taping techniques in increasing navicular height and anti-pronation immediately post taping and exercise for a period of time. The results of the meta-analysis could provide better understanding on their efficacies, may improve the clinical outcome and provide a practical alternative for clinical taping intervention with pes planus.

## Methods

### Search strategy

This research was registered on "International prospective register of systematic reviews" (CRD:42020184310) from University of York. An extensive literature search for all quasi-randomised and clinical controlled trials examining efficacy of taping was conducted by two independent reviewers on the following databases (updated to March, 2020): Web of Science, Pubmed, EBSCO, CNKI and Cochrane Library. The keywords and phrases used in the online search included 'tape', 'Athletic Tape', 'taping' 'strap', 'arch', 'navicular', 'Subtalar Joint', 'Foot Joints', 'Foot', 'Talus', 'Flatfoot', 'pes planus', 'Pronation' and 'pronated'.

The search terms were based on modified PICO principles (princles following patients, intervention, comparison and outcomes) to search through the databases above to access all the essential articles. The detailed electronic search strategies are shown in the Appendix. Reference lists from published papers were also reviewed to identify any other relevant studies not cited in the online databases.

### Selection

Articles were included in the systematic review if they met the following criteria:

1. Quasi-randomised or clinical controlled trials published in peer-reviewed journals;

2. Female and male subjects (18–40 years) who were diagnosed as having musculoskeletal conditions potentially related to arch collapse and otherwise healthy, such as pronated foot or navicular drop distance is greater than 8 mm or observed a lower medial longitudinal arch height (e.g. excessive pronation) during the stance phase of walking, and/or FPI score is greater than or equal to 6, and/or RCSP is greater than 4°[19,22,27,28].

3. Comparing the taping techniques with baseline or a no-intervention control group (or sham tape);

4. Studies included one of the following outcome measurements of foot arch collapse: navicular height, navicular drop distance, foot posture index (FPI), pronation angle and resting calcaneal stance position (RCSP);

5. Measuring the foot arch collapse during activities of walking or jogging or running;

6. The full text was written in English or Chinese.

   The exclusion criteria were listed as follows:

1. Studies not meeting the minimum requirements of an experimental study design (i.e. case report or review article);

2. Studies on subjects with any neurological diseases or ankle and foot injury or pathology in the past 6 months;

3. Studies that did not compare the effect of intervention immediately after tape or after a period of exercise with baseline.

Two independent reviewers screened the title and abstract of each article to decide whether it should be selected based on the preset criteria. For the actual review, two criteria were used to determine whether the article should be included in the analysis: (1) tabular means and standard deviation (SD) were available for baseline, post-tape and taping after a period of exercise. In articles where data reporting was incomplete, the authors of the articles would be contacted for clarifications. If the result data was presented graphically, then we used Engauge Digitiser 12.1 software to quantify. (2) The quality of the study was satisfactory according to the Cochrane scales.

The outcome used in this study was restricted to the common outcomes, such as change in navicular height (NH), navicular drop distance (NDD), Foot Posture Index (FPI), resting calcaneal stance position (RCSP) and pronation angle, to accommodate studies with different taping methods. The FPI is a six-item clinical assessment tool used to evaluate foot posture, with an acceptable validity and good intra-rater reliability (ICC = 0.893–0.958) [29–31]. The FPI evaluates the multi-segmental nature of foot posture in all three planes8, and it does not require the use of specialised equipment. Each item of the FPI is scored between −2 and +2. Thus, a total between −12 (highly supinated) and +12 (highly pronated) is obtained. Items include talar head palpation, curves above and below the lateral malleoli, calcaneal angle, talo-navicular bulge, medial longitudinal arch and forefoot to hindfoot alignment. Participants were assessed whilst in a relaxed standing position. FPI score >5 is defined as foot pronation, pes planus feet independently demonstrated over-pronation [32]. An RCSP of over 4˚ was essential for inclusion as this is classified as excessively pronated [33]. Pronation angle was taken as the amount of pronation from heel contact to maximum pronation angle. Excessive pronation was defined as navicular drop distance over10 mm [34]. These dynamic and static outcome measurements were commonly used for foot arch collapse. Even though other outcome variables, such as change in COP and EMG, were clinically important, extraction of these data for this meta-analysis was not done in the absence of a common definition of these variables among studies. If the outcome indicators were presented as pictures, then Engauge Digitiser 12.1 was used.

## Quality assessment

The quality of the studies included in this analysis was evaluated by the Cochrane scale by two independent reviewers. The Cochrane bias risk assessment tool mainly covers seven aspects: random sequence generation and allocation hiding, blinding of researchers and subjects, blind evaluation of research outcome, completeness of outcome data, selective reporting of research results and other sources of bias. For each item, the judgment results of 'low risk bias', 'high risk bias' and 'unclear' were made according to the bias risk assessment criteria. Disagreements about the Cochrane criterion score were resolved by a third reviewer.

## Statistical methods

All of the data from included studies were collected and expressed as mean ± SD for the control and experimental groups. The pooled mean difference of treatment effects was calculated and illustrated by forest plots with Review Manager version 5.3 (The Nordic Cochrane Centre, The Cochrane Collaboration, Copenhagen, Denmark). The heterogeneity of the included

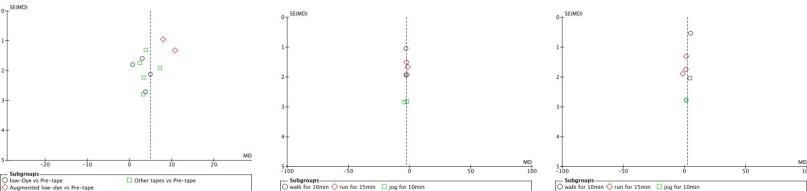

**Fig 1. Funnel plots of included studies.** (A) Navicular height Post tape VS pre-tape. (B) Navicular height post exercise VS immediately post tape. (C) Navicular height post exercise VS pre-tape.

studies was tested by $I^2$ index. The $I^2$ index was selected instead of the Q statistics because the latter only provided information regarding the existence of heterogeneity but did not quantify the extent of such heterogeneity. $I^2$ index of ≤50% demonstrated low heterogeneity and supported a fixed-effect model in the meta-analysis. On the contrary, random effect model in the meta-analysis should be used when the $I^2$ index was >50%, which was suggestive of a moderate to high heterogeneity of the studies [35].

Bias funnel plots (Fig 1) were used to illustrate the relationship between intervention effect and SE in each intervention. Publication bias assessment was tested by the Egger's regression intercept using Comprehensive Meta-analysis version 2 (Biostat, Englewood, New Jersey, USA) [36]. A p value of <0.1 (two tailed) in the test indicated the presence of publication bias.

## Results

After the initial electronic search was completed, 93 articles were identified (Fig 2). Among the 93 studies, 64 had not examined the target outcome measurements in this meta-analysis and were excluded. Four were meta-analysis or systematic review, and they were also excluded. Of the remaining 25 reports, 7 studies had no full text and were also excluded. Another 2 studies were rejected due to incomplete results reported (by arch height index or arch index) and because the authors of that study did not reply to our inquiry. The remaining 16 studies were included in this meta-analysis (Table 1). Among the 16 studies, 8 studied navicular height, 2 studied navicular

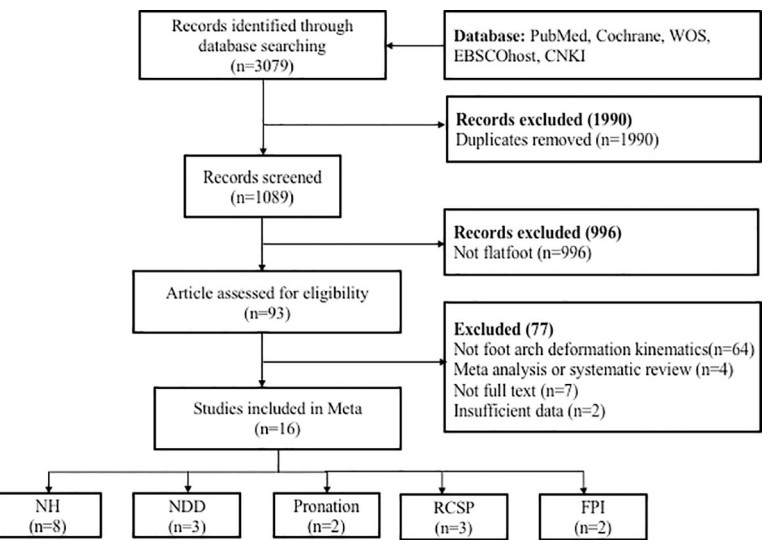

**Fig 2. Flow chart.** Navicular Height; NDT: Navicular Drop Distance; RCSP: resting calcaneal stance position; FPI: Foot posture index.

**Table 1. The summary of included studies examining the efficacy of kinesio tape.**

| Study | N | Taping materials | Taping techniques | Testing condition | MD | SD | SE | MD (95%CI) |
|---|---|---|---|---|---|---|---|---|
| **Outcome 1: Navicular height** | | | | | | | | |
| Holmes CF, 2020 [21] | 40 | Rigid tape | Modified low-Dye | Post tape | 7.2 | 3.78 | 1.94 | 3.42 to 10.98 |
| | | | | walk 10min | -2.8 | 3.8 | 1.95 | -6.60 to 1.00 |
| Larson T J, 2019 [18] | 25 | 1½ inch Johnson & Johnson Coach Athletic Tape, 1-inch Zonas Porous Tape (AQ1), and Colorless Tuf Skin. | low-Dye | Post tape | 4.9 | 4.16 | 2.04 | 0.74 to 9.06 |
| | | | Fan-arch support | Post tape | 3.32 | 4.38 | 2.09 | -1.06 to 7.70 |
| Newell T, 2015 [19] | 25 | White cloth tape (Coach; Johnson & Johnson Consumer Products, Inc, New BRunswick, NJ); 1.5-in (3.81-cm) | low-Dye | Post tape | 0.7 | 3.52 | 1.88 | -2.82 to 4.22 |
| | | | | Run 15min | -2 | 3.77 | 1.94 | -5.77 to 1.77 |
| | 25 | Elastic tape (Elastikon; Johnson & Johnson Consumer Products, Inc). 2 in (5.08 cm) wide | Navicular Sling | Post tape | 2.4 | 3.41 | 1.85 | -1.01 to 5.81 |
| | | | | Run 15min | -1.4 | 3.27 | 1.81 | -4.67 to 1.87 |
| Franettovich M, 2008 [23] | 5 | A rigid sports tape (38-mm zinc oxide adhesive, Leukosport, BDF) | Augmented low-Dye | Post tape | 8 | 1.87 | 1.37 | 6.13 to 9.87 |
| | | | | walk 10min | -3 | 2.05 | 1.43 | -5.05 to -0.95 |
| Del Rossi G, 2004 [24] | 8 | A rigid sports tape (38-mm zinc oxide adhesive, Leukosport, BDF) | Double-X | Post tape | 3.83 | 2.58 | 1.61 | 1.25 to 6.41 |
| | | | Double-X | Run 15min | -2.54 | 2.94 | 1.71 | -5.48 to 0.40 |
| Whitaker JM, 2003 [27] | 22 | Single-sided adhesive athletic tape strips (1 and 3 inches) | low-Dye | Post tape | 3.08 | 3.12 | 1.77 | -0.04 to 6.20 |
| Vicenzino B, 2000 [37] | 14 | Rigid sports tape (38 mm) with a zinc oxide adhesive | Augmented low-Dye | Post tape | 10.8 | 2.61 | 1.62 | 8.19 to 13.41 |
| Ator R, 1991 [38] | 10 | One-inch adhesive tape | low-Dye | Post tape | 3.8 | 5.3 | 2.3 | -1.50 to 9.10 |
| | | | | jog 10min | -2.35 | 5.51 | 2.35 | -7.86 to 3.16 |
| | | | Double- X | Post tape | 3.25 | 5.48 | 2.34 | -2.23 to 8.73 |
| | | | | jog 10min | -4.3 | 5.57 | 2.36 | -9.87 to 1.27 |
| **Outcome 2: Resting Calcaneal Stance Position** | | | | | | | | |
| Harradine P, 2001 [39] | 7 | 25 mm wide rigid zinc oxide 'Leukoplast' tape. | low-Dye | Post tape | -2.1 | 2.85 | 1.69 | -4.95 to 0.75 |
| LeeSojung, 2017 [40] | 13 | Original Kinesio tape (Nasara, Japan) Materials: cotton 96% and spandex 4% Length: 5 m, Width: 5 cm | Kinesiotape | Post tape | 5.77 | 2.15 | 1.47 | 3.62 to 7.92 |
| Whitaker JM, 2003 [27] | 22 | Single-sided adhesive athletic tape strips (1 and 3 inches) | low-Dye | Post tape | -4.59 | 1.65 | 1.28 | -6.24 to -2.94 |
| **Outcome 3: Pronation angle** | | | | | | | | |
| O'Sullivan K, 2008 [41] | 28 | A standard LD taping technique using rigid 3.8 cm wide zinc oxide tape (Leukotape) | low-Dye | Post tape | -1.2 | 5.01 | 2.24 | -6.21 to 3.81 |
| Harradine P, 2001 [39] | 7 | 25 mm wide rigid zinc oxide 'Leukoplast' tape. | low-Dye | Post tape | 1.39 | 2.49 | 1.58 | -1.10 to 3.88 |
| **Outcome 4: Navicular drop distance** | | | | | | | | |

*(Continued)*

**Table 1.** (Continued)

| Study | N | Taping materials | Taping techniques | Testing condition | MD | SD | SE | MD (95%CI) |
|---|---|---|---|---|---|---|---|---|
| Siu WS, 2020 [14] | 9 | Kinesiotape | unnamed | Post tape | -5.55 | 1.71 | 1.85 | -7.26 to -3.84 |
| Kim T, 2017 [42] | 24 | Kinesiotape | unnamed | Post tape | -0.09 | 1.57 | 1.25 | -1.66 to 1.48 |
| Prusak KM, 2014 [43] | 20 | Leukotape Sports Tape (BSN Medical Inc, Charlotte, North Carolina),a rayon-backed tape with an aggressive zinc oxide adhesive | Augmented low-Dye | Post tape | -0.86 | 1.11 | 1.05 | -1.97 to 0.25 |
| | | | Antipronation spiral stirrup (APSS) | Post tape | -2.72 | 1.28 | 1.13 | -4.00 to -1.44 |
| **Outcome 5: Foot Posture Index** | | | | | | | | |
| Aguilar MB, 2016 [44] | 34 | Standard 5 cm Black Irisana©tape | low-Dye | Post tape | 0.1 | 0.81 | 0.9 | -0.71 to 0.91 |
| | | | | Run 45min-1h | -0.59 | 1.13 | 1.06 | -1.72 to 0.54 |
| | | | | Run 45min-1h | -2.95 | 1.04 | 1.02 | -3.99 to -1.91 |
| Luque-Suarez A, 2014 [28] | 73 | kinesiotape, Standard 5-cm blue Cure ©tape | Kinesiotape | Post tape | -0.07 | 0.79 | 0.89 | -0.86 to 0.72 |
| | | | | After 45min-1h | -1 | 1.31 | 1.14 | -2.31 to 0.31 |
| | | | | After 45min-1h | -3.6 | 0.73 | 0.85 | -4.33 to -2.87 |

Note: N: Participants; MD: Mean difference; SD: Standard Deviation; SE: Standard Error.

drop distance, 2 studied pronation angle, 3 studied RCSP and 2 studied FPI. Two studies had examined more than one outcome above. Thus, the total study number exceeded 16. The total number of subjects involved in these studies was 486. The characteristics of the subjects are presented in Table 2. The Cochrane scores of the selected studies are listed in Table 3.

High heterogeneity between datasets was found in studies with different interventions, and the $I^2$ value was 75% for navicular height immediately after tape, 89% for navicular drop distance, 93% for FPI and 96% for RCSP. However, the value was 0% for navicular height after exercise and pronation. In other words, variability in treatment effect estimates was not only due to sampling error within studies. Thus, group navicular height immediately after tape, navicular drop distance, FPI and RCSP used random effect model; the left group used fixed model.

The bias funnel plots of these studies are shown in Fig 1. As shown in Fig 1, the funnel plot was slightly asymmetric across navicular height, navicular drop distance, FPI and RCSP outcome studies. By contrast, the plot was symmetrical in navicular height post exercise and pronation angle outcome studies. These results suggested that publication bias was not present in the navicular height post exercise and pronation angle outcome studies, but marginal publication bias existed for navicular height, navicular drop distance, FPI and RCSP outcome studies.

The weighted mean differences in kinematic parameters and pooled variance data for individual antipronation interventions were represented by forest plots (Fig 3).

## Effect of taping on navicular height and drop distance immediately post tape compared with pre-tape

All taping techniques included in our study (ALD, MLD, low-Dye, Navicular sling, Double X and Fan-arch support) significantly increased navicular height immediately post tape

**Table 2. Characteristics of participants in the included studies.**

| Study | Age | Gender | Gender control | Atheletic | Foot-type control | Other characteristics | Exercise intervention | Speed | Duration |
|---|---|---|---|---|---|---|---|---|---|
| Holmes CF, 2020 [21] | 27(SD NR) | mixed | N | N | Y | Difference between STJN and the relaxed position was a positive number | Walking | NR | 10min |
| Larson T J, 2019 [18] | 19.8±1.04 | Mixed | N | Y | Y | Navicular drop distance ≥ 10 mm | Walking and running | Faster than jog but slower than a full sprint and to complete the walking laps at a brisk but comfortable pace. | 20min |
| Newell T, 2015 [19] | 20.0±1.0 | Mixed | N | N | Y | Navicular drop distance ≥ 8 mm | Jogging | Self-selected speed, 8.1 ±1.3 km/h | 15min |
| Franettovich M, 2008 [23] | 36.4 ± 7.5 | Mixed | N | N | Y | A lower medial longitudinal arch height (e.g., excessive pronation) during the stance phase of walking | Walking | Began at a selfselected speed and then gradually increased to the test speed of 4.5 km/h | 10min |
| Del Rossi G, 2004 [24] | F:19.4 ±0.5 M:23.3±1.2 | Mixed | N | N | Y | Navicular drop distance≥ 10 mm | Treadmill running | Self-selected speed | 30min |
| Whitaker JM, 2003 [27] | NR | NR | N | N | Y | A certain degree of excessive flexible subtalar joint pronation and a certain degree of medial column collapse. | N | N | N |
| Vicenzino B, 2000 [37] | 23.8 ±3.5 | Mixed | N | N | Y | Navicular drop distance> 10 mm | Jogging | NR | 20min |
| Ator R, 1991 [38] | 22.8(SD NR) | F | Y | N | NR | NR | Jogging | NR | 10 min |
| Kim T, 2017 [42] | 21(SD NR) | Mixed | N | Y | Y | Navicular drop distance≥ 10 mm | Walking or jogging | Self-selected speed | 6min |
| Prusak KM, 2014 [43] | 21.9 ±2.27 | Mixed | N | N | Y | Navicular drop distance≥ 10 mm | N | 3 mph at 0% grade | 15min |
| Siu WS, 2020 [14] | 21.11± 1.27 | Mixed | N | NR | Y | Navicular drop distance> 10 mm | Treadmill running | NR | 20min |
| Aguilar MB, 2016 [44] | 28.9±6.0 | Mixed | N | Y, | Y | 6<FPI<12 | Treadmill running | 12 km/h | 45min |
| Luque-Suarez A, 2014 [28] | 25±6.5 | Mixed | N | NR | Y | 6<FPI<12 | N | N | N |
| Harradine P, 2001 [39] | NR | Mixed | N | NR | Y | RCSP > 4˚ | Walking | 4.8 ± 0.75 k.p.h | 30min |
| LeeSojung, 2017 [40] | 24.8 ± 3.0 | F | Y | NR | Y | Flat-footed | Walking | 1.3 m/s | NR |
| O'Sullivan K, 2008 [41] | 22.1 ± 5 | Mixed | N | N | Y | Navicular drop distance > 10 mm | Walking | Self-selected speed | 10-metre walkway |

Note: F, female; M, male; N, no; NR, not reported; Y, yes; RCSP: resting calcaneal stance position; FPI: Foot posture index.

compared with baseline (p<0.001). Among them, the increase amplitude of navicular height in ALD was the highest (mean difference = 10.8 mm, 95% CI = 8.19–13.41, Z = 6.64), that in MLD was second, and it was followed by those in other taping techniques (mean difference = 2.86 mm; 95% CI = 0.96–4.76; Z = 2.95), such as Navicular sling [44], Fan-arch support and Double X. The least effective taping technique was low-Dye, and the overall difference between the most and least effective taping techniques was less than 8 mm.

Navicular drop distance during weight bearing was calculated as the navicular height of sitting minus that of standing [27]. This result was consistent with that in navicular height.

**Table 3. Cochrane criterion score of included studies.**

| Study | Adequate sequence generation | Allocation concealment | Blinding of participants and personnel | Blinding of outcome assessment | Incomplete outcome data addressed | Free of other bias | Others |
|---|---|---|---|---|---|---|---|
| Holmes CF, 2020 [21] | H | L | L | L | L | L | L |
| Larson T J, 2019 [18] | L | H | L | L | L | L | L |
| Siu WS, 2020 [14] | N | N | L | L | L | L | L |
| LeeSojung, 2017 [40] | N | N | L | L | L | L | L |
| Kim T, 2017 [42] | L | N | L | N | L | L | L |
| Aguilar MB, 2016 [44] | L | L | L | L | L | L | L |
| Newell T, 2015 [19] | L | N | L | L | L | L | L |
| Prusak KM, 2014 [43] | L | N | L | L | L | L | L |
| Luque-Suarez A, 2014 [28] | N | L | L | L | L | L | L |
| O'Sullivan K, 2008 [41] | H | N | L | L | N | L | L |
| Franettovich M, 2008 [23] | H | N | L | L | L | L | L |
| Del Rossi G, 2004 [24] | L | N | L | L | L | L | L |
| Whitaker JM, 2003 [27] | L | N | L | L | L | L | L |
| Harradine P, 2001 [39] | N | N | L | L | N | L | L |
| Vicenzino B, 2000 [37] | L | N | L | L | L | L | L |
| Ator R, 1991 [38] | L | N | L | L | L | L | L |

Note: H: High; L: Low; N: Unclear.

Specifically, a reduction in navicular drop distance was observed immediately after tape (mean difference = −2.27 mm, 95% CI = −4.37–0.16, Z = 2.11, p = 0.05).

### Effect of taping on navicular height post exercise compared with immediately post tape and baseline

The results revealed that taping techniques had a significant weighted mean decrease after exercise in navicular height compared with immediately post tape (mean difference = −2.59 mm, 95% CI = −3.83 to −1.35, Z = 4.10, p<0.0001). After further sensi analysis, we found that this difference came from walking for 10 min (mean difference = −2.96 mm, 95% CI = −4.76 to −1.15, Z = 3.21, p = 0.001) and running for 15 min (mean difference = −2.02 mm, 95% CI = −3.91 to −0.13, Z = 2.10, p = 0.04). Navicular height after running for 15 min and walking for 10 min were significantly lower than those immediately post tape.

Navicular height significantly increased after exercise compared with baseline (mean difference = 3.78 mm, 95% CI = 2.91–4.64, Z = 8.56, p<0.001), but the heterogeneity among studies included was high ($I^2$ = 70%). Although greater reduction in navicular height was observed for group walking for 10 min than running for 15 min group compared with immediately post tape, navicular height after running for 15 min decreased nearly to baseline (mean difference = 0.60 mm, 95% CI = −1.20 to 2.39, Z = 0.65, p = 0.52), navicular height after walking for

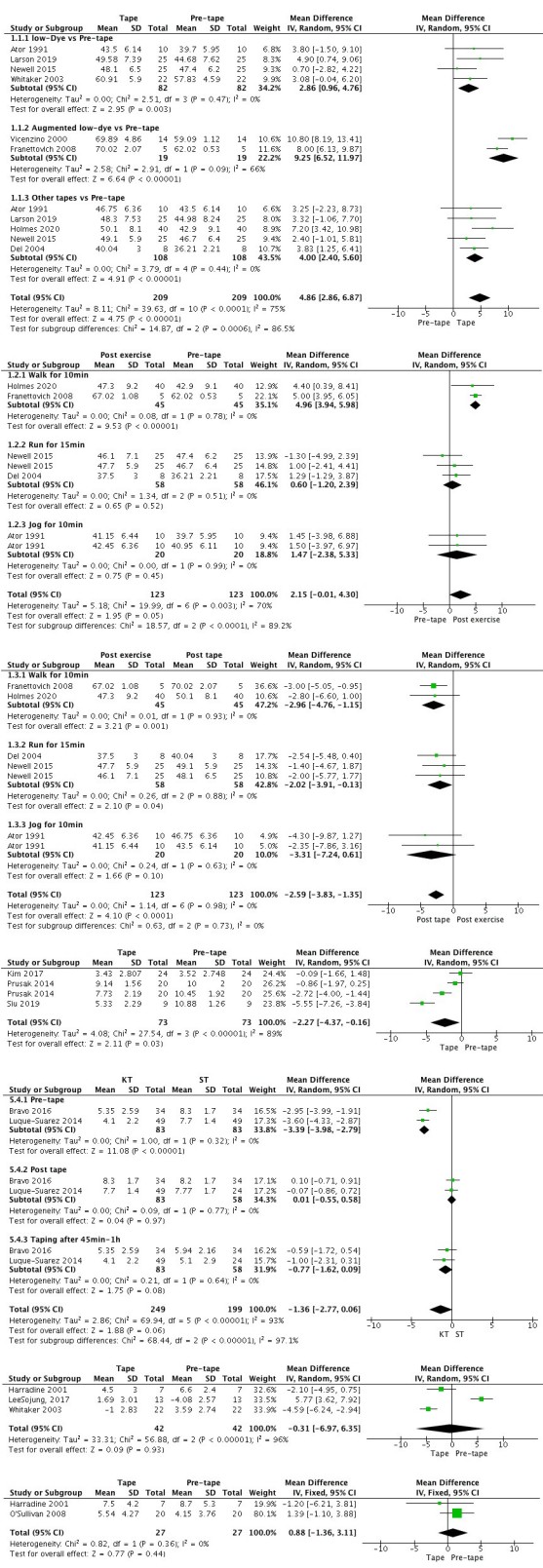

**Fig 3.** (A) Forest plot for efficacy of tape on the navicular height post-tape compared with pre-tape; (B) Forest plot for efficacy of tape on the navicular height post exercise compared with post-tape; (C) Forest plot for efficacy of tape on the navicular height post-tape compared with pre-tape; (D) Forest plot for acute and post exercise efficacy of kinesio

tape on FPI compared with sham tape; (E) Forest plot for efficacy of tape on NDD post-tape compared with pre-tape; (F) Forest plot for efficacy of tape on RCSP post-tape compared with pre-tape; (G) Forest plot for efficacy of tape on pronation angle post-tape compared with pre-tape.

10 min was much higher than baseline (mean difference = 4.96 mm, 95% CI = 3.94–5.98, Z = 9.53, p<0.001) and MLD technique (MD = −2.80 mm) had a smaller decrease than ALD technique (MD = −3.00 mm) after walking for 10 min. This finding indicated that ALD was better than MLD in immediate improvement in navicular height, whilst MLD was better than ALD for a period of exercise.

By contrast, no significant change in navicular height was observed after jogging for 10 min compared with immediately post tape (p = 0.10) and pre-tape (p = 0.45). Currently, studies on navicular height after jogging are limited. Thus, the actual results might be amplified when the statistics were pooled.

## Effect of taping on FPI, RCSP and pronation angle immediately post tape compared with pre-tape

No significant difference in FPI was observed when compared with sham tape (mean difference = −1.36 mm, 95% CI = −2.77 to 0.06, Z = 1.88, p = 0.06). According to testing condition, this group was further subdivided into pre-exercise after 45 min to 1 h. Although no significant changes over the two time points were found, significant reduction was observed for 45 min to 1 h later when compared with pre-tape (barefoot). Therefore, kinesio taping techniques tended to reduce arch deformation and make foot approach neutral position.

No difference was observed between the pronation angle of the foot after a period of exercise and baseline (p = 0.44). In addition, no significant changes were observed for RCSP immediately after tape (mean difference = −0.31 mm, 95% CI = −6.97 to 6.35, Z = 0.09, p = 0.93). However, RCSP after low-Dye taping was closer to a neutral position.

## Discussion

This meta-analysis compared the efficacy of various athletic taping techniques for controlling foot arch deformation. The findings revealed that ALD was the most effective taping technique for controlling foot arch collapse immediately post tape compared with baseline. Moreover, MLD was better than ALD in maintaining immediate navicular height after walking for 10 min. However, the overall difference in foot arch deformation among the taping techniques after exercise was less than 11 mm in the current study. Currently, no definitive data are available on a clinically meaningful increase in navicular height. A case series study on seven high school athletes with a lower limb overuse injury reported reductions in pain by 23%–91% with ALD taping for 4 weeks, it also caused increase in navicular height as small as 0.1–0.3 cm [45]. Thus, the small change in our study might have clinical significance for individuals with pes planus.

## Immediate increase in navicular height, FPI, pronation angle and RCSP

We found that all taping techniques included in this meta-analysis immediately improved navicular height in adults with pes planus. ALD performed better, and it was followed by MLD. Kim and Park assessed acute changes in normalised navicular height in 24 healthy elite athletes using 3D motion analysis after tape during sitting, standing, walking and jogging [42]. A positive effect of rigid tape was reported. It resulted in a significant larger increase in

normalised navicular height than kinesio tape. Therefore, rigid taping techniques might work for healthy participants and adults with pes planus [46].

In previous studies on ALD taping techniques, the researchers found a higher weighted mean difference in navicular height among participants with a change in vertical navicular height of greater than 10 mm (mean difference = 10.8 mm 95% CI = 8.19–13.41, Z = 6.64, p<0.001) than that among participants with a lower medial longitudinal arch height (e.g. excessive pronation) during the stance phase of walking in another study (mean difference = 8.00 mm, 95% CI = 6.52–11.97, Z = 6.64, p<0.001) [23,37]. We supposed that the discrepancies might be caused by the different inclusion criteria of the participants. Specifically, the strict inclusion criteria of former study had a navicular drop distance greater than 10 mm, whilst another study only had a lower medial longitudinal arch. As a result, the navicular height of the former changed more significantly.

The changes in RCSP were insignificant among low-Dye taping techniques. The result was consistent with a previous systematic review [47]. Pronation is a normal part of the stance phase of gait, it's measured as the angle from heel contact to maximum pronation angle during exercise, excessive pronation in the rearfoot remains pronated beyond the midstance phase of gait [48]. However, the findings on pronation angle were contradictory to those in another meta-analysis [16]. The reason for this difference was that the previous meta-analysis used rearfoot eversion instead of the pronation angle indicator [16]. In addition, the results showed that kinesio tape had no significant effects on FPI in healthy adults, which was in disagreement with the result of a randomly Double-blinded study [44]. Although a score closer to neutral was observed in kinesio group, significant changes were found in sham tape group as well. This result might be due to the self-perception of kinesio taping techniques. Although this influence might occur when applied with no tension, an unconscious mechanism of correction could be induced in adults with ankle instability. Thus, the patient's belief in the taping technique was thought to produce a placebo effect [49]. Therefore, the observed significant changes were caused by the patient's self-perception. Accordingly, this result was essentially the same as our findings.

## Post-exercise change in navicular height

The results from previous research showed that exercise could cause athletic tape to lose some of its restricting properties [45,50], which was consistent with our findings. In previous studies, Navicular height in adults with pes planus significantly decreased after walking for 10 min [21,23] (p = 0.001) and running for 15 min [19,24] (p = 0.04) compared with immediately post taping. Navicular height was still higher after walking for 10 min than baseline, but it returned to baseline after running for 15 min. Similar to our findings, a recent study investigated kinesio tape on navicular drop distance among individuals with pes planus [14]. The results showed that kinesio tape exhibited a significant reduction in navicular drop distance (mean difference = 5.56 mm) immediately, but this effect was not sustained after running. One common explanation was that it lost its effectiveness after exercise because of diminished adherence to the skin [21,38]. The tape was applied directly to the skin. Thus, perspiration during activity could have caused a reduction in tape adherence. Moreover, the taping technique became less effective due to the loss of tensile strength quality when skin moved together with running [21]. We supposed that exercise intensity and duration were responsible for the maintained immediate effect difference between running and walking, but this deduction needs more studies. MLD was better than ALD in maintaining immediate navicular height after walking for 10 min. This performance might be due to that MLD technique more deeply corrected the excessive pronation of midtarsal joint (subtalar joint) in adults with pes planus than ALD [51].

In addition, navicular height insignificantly changed after jogging for 10 min compared with baseline (p = 0.45) and immediately post tape (p = 0.10) [38]. The reason for this result was that foot was in contact with the ground for a longer time when jogging, which caused the adherence material to fail due to repeated friction [52]. The Double X taping technique provided poor leverage to support foot arch [38,50]. The inconsistent results recommended that more RCTs are needed in the future. Among eight studies involving taping techniques, two had adopted walking test, two running test and one jogging test. The lesser amount of navicular height post walking might overestimate the treatment effects in this group, and this condition led to the significant findings.

Apart from Navicular sling taping technique using elastic tape [19], all studies included in our meta-analysis used rigid white tape. White tape is a non-elastic, cloth rigid tape and has been widely used to restrict joint range of motion; thus, it works to prevent and treat lower limb injuries [53]. We found that Navicular sling taping technique (elastic tape) was better in lifting navicular height than low-Dye technique (white tape) immediately post tape. This effect could be maintained after walking for 10 min. The type of tape that was applied and the basic construction of the taping technique were responsible for the difference. low-Dye is one of the most common techniques among physical therapy methods [18,19,27,38], and it consists of an anchor strip and a series of transverse strips. The anchor strip, starting from the fifth metatarsal head to calcaneal tubercle and carrying over to the first metatarsal head, provided the base for attachment of the transverse strips. The transverse strips running from lateral to medial of the anchor strip under the plantar surface of the foot would counteract the pronation movement. Firstly, compared with low-Dye technique, Navicular-sling taping technique applied a thicker and stronger elastic kinesio tape that had better adhesive qualities [19]. The researchers observed that the low-Dye taping technique pulled away from the skin as participants ran during testing procedures, but the Navicular sling technique did not. Secondly, the Navicular sling used an 8-shaped tape to the ankle, which was probably better in foot arch contraction [19]. Previous studies have also found that more complex strapping techniques could maintain effectiveness for a longer time [38,54]. Researchers have also reported that techniques such as the Double X and the modified low-Dye technique with additional 'reverse-6' strips wrapped further up to the ankle provided longer support than traditional low-Dye techniques [38,50,54].

## Neurophysiological mechanics among different taping techniques

The activity of tibialis anterior was significantly greater in the pes planus than in the neutral foot during sitting short-foot exercises and standing [55,56]. ALD taping has been proved to decrease activities of the tibialis anterior, tibialis posterior and peroneus longus during walking [2]. Similarly, Franettovich MM et al. found that the peak myoelectricity of the tibialis posterior, peroneus longus and tibial anterior reduced by 33.1%, 29.4% and 13.1%, respectively, when MLD taping technique was applied compared with barefoot walking [25]. The study by Franettovich et al. also found that the myoelectric activity of the external muscles around the ankle decreased when MLD taping technique was used in adults with lower arches [23]. The aforementioned studies show that ALD and MLD taping techniques play a role in adjusting nerve electrophysiological activity during short-term walking. This performance might be due to that the athletic white tape gives rigid and static correction of the pes planus. Therefore, the lower leg muscles require less effort to maintain the navicular height. However, in a longer-term intervention programme, Franettovichet al. assessed lower limb muscle activity with EMG in 28 females with ALD tape during walking [26]. The results showed that continual use of ALD tape for 12 days might increase arch height but did not change lower limb muscle

recruitment. This finding demonstrated that these observed changes probably represented natural variation in muscle activation patterns rather than a result of the taping technique intervention. An alternative possibility is that a longer intervention period (exceeding 12 days) is required for adaptations in neuromotor control to occur [26]. Obviously, the physical support provided by the tape when it is sustained over a period of time may result in soft tissue adaptations [26]. In future research, we could further explore the adaptive changes in the plantar fascia, ligaments of passive subsystem, intrinsic and extrinsic foot muscles of active subsystem and musculotendinous receptors in neural subsystem in addition to investigating the effect of taping techniques on the navicular height in the passive subsystem of the foot core system [57]. Different from ALD technique, the tibialis anterior was observed to be more activated with kinesio tape than no tape during 8–10 min treadmill running in adults with pes planus [14]. The increase in muscle activity of tibialis anterior may demonstrate the dynamic stabilising effect on the foot arch during weight-bearing exercise with taping techniques [14]. We speculated that tape materials and taping techniques were responsible for the difference. Further research could focus on controlling the independent variable of one of the taping techniques above to explore the different taping techniques with same taping material in adults with pes planus.

Our meta-analysis found that ALD technique had the best effect in lifting navicular height immediately. In addition, the MLD technique was better than ALD taping technique after walking for 10 min. After running for 15 min, the elastic tape used by the Navicular sling technique was better than that by the low-Dye white taping technique in maintaining navicular height. We supposed that ALD technique could be improved in the future. Specifically, we would investigate whether using three reverse sixes and two calcaneal slings in elastic kinesio tape instead of white athletic tape with the first low-Dye technique would have a better effect on navicular height than traditional ALD technique in adults with pes planus immediately post tape and after a period of running.

MLD taping technique was the most effective taping technique in controlling foot arch collapse after walking for 10 min, but it was measured with taping on the skin of adults with pes planus. The long-term effect of improving the arch collapse after falling off remains unknown. In the meantime, whether MLD techniques could maintain the navicular height after a period of running is still unclear. This topic needs further investigation.

## Limitations

This study focused on the effects of various taping techniques on passive subsystem of the foot arch due to the limitations of meta-analysis combined with statistics calculation. Further systematic review could pay attention to the adaptive changes in neural subsystem and active subsystem of the foot arch.

Compared with other conservative interventions, various and complex taping techniques are more dependent on the application technique than foot orthoses and arch support insoles. Thus, a professional physical therapist is required to evaluate the operation in clinical practice every time. Accordingly, this technique seems inconvenient for users. Taping techniques are also unsuitable for people with skin allergies and may cause new overuse injuries. Therefore, if one expects a long-term reduction of subtalar joint pronation during running, then foot orthotics (e.g. motion control footwear) could be used as an alternative [40,51,58].

## Conclusion

ALD was the most effective taping technique for controlling foot arch collapse immediately post tape compared with baseline, and it was followed by MLD. MLD could possibly

performed better than ALD in maintaining immediate navicular height after walking for 10 min, which needs further investigation in the future. Low-Dye could make resting calcaneal stance position closer to neutral position. Although positive effects of Navicular sling, low-Dye and Double X interventions on navicular height in adults with pes planus were observed, they could not maintain this effect during running. For adults with pes planus, the results of this study recommend that the small increase in navicular height created by taping can be clinically significant. However, practitioners should practice caution when using the technique to improve arch deformation during exercise.

## Supporting information

**S1 Checklist.**
(DOC)

**S1 Table. Abbreviation key list.**
(DOCX)

**S1 Data. Raw and converted data, characteristics of participants from included studies.**
SD: Standard deviation. SE: Standard error. Mean difference(95%CI): 95% Confidence Interval. NR: Not reported. F: Female; Mixed: Male and female.
(XLSX)

**S2 Data. Searching terms in database Pubmed, Web of Science.**
(XLSX)

## Acknowledgments

We would like to express our gratitude to those who helped us during the process of this article. I gratefully acknowledge the help of my supervisors Xiaoyue Hu and Lin Wang, who have offered me valuable suggestions in this article.

## Author Contributions

**Resources:** Yanwei You, Jiajia Li.

**Supervision:** Xiaoyue Hu.

**Validation:** Yanwei You, Jiajia Li.

**Writing – original draft:** Meihua Tang.

**Writing – review & editing:** Lin Wang.

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
