## [Decision Letter · Decision Letter 0]

8 Apr 2021

PONE-D-20-37311

Effects of taping techniques on arch deformation in adults with pes planus: A meta-analysis

PLOS ONE

Dear Dr. Wang,

Thank you for submitting your manuscript to PLOS ONE. After careful consideration, we feel that it has merit but does not fully meet PLOS ONE’s publication criteria as it currently stands. Therefore, we invite you to submit a revised version of the manuscript that addresses the points raised during the review process.

We look forward to receiving your revised manuscript.

Kind regards,

Slavko Rogan

Academic Editor

PLOS ONE

Journal Requirements:

'No, the funders had no role in study design, data collection and analysis, decision to publish, or preparation of the manuscript.'

4. Please amend the manuscript submission data (via Edit Submission) to include authors Jiajia Li and Yanwei You.

5. We note you have included a table to which you do not refer in the text of your manuscript. Please ensure that you refer to Table 4 in your text; if accepted, production will need this reference to link the reader to the Table.

6. Please include captions for your Supporting Information files at the end of your manuscript, and update any in-text citations to match accordingly. Please see our Supporting Information guidelines for more information: http://journals.plos.org/plosone/s/supporting-information

Reviewers' comments:

Reviewer's Responses to Questions

**Comments to the Author**

1. Is the manuscript technically sound, and do the data support the conclusions?

Reviewer #1: Yes

Reviewer #2: Yes

2. Has the statistical analysis been performed appropriately and rigorously? 

Reviewer #1: Yes

Reviewer #2: Yes

3. Have the authors made all data underlying the findings in their manuscript fully available?

Reviewer #1: Yes

Reviewer #2: Yes

4. Is the manuscript presented in an intelligible fashion and written in standard English?

Reviewer #1: Yes

Reviewer #2: Yes

5. Review Comments to the Author

Reviewer #1: Thank you authors for manuscuripts. As a result of the evaluation;

1. The authors has made objective clearly stated and clinically relevant and focused study question included

2. Authors has made Comprehensive literature search conducted and searched information sources listed (ie, PubMed, Cochrane database). Terms used for electronic literature search provided, author may be add terms ‘Pes Planus’

3. Review included at least two reviewers should search sources for articles relevant to the meta-analysis, and the keywords used in the online searches should be provided in the article.

4. Inclusion and exclusion criteria have been defined for studies included in the meta-analysis.

5. The meta-analysis have been provide a table outlining the features of the studies,

6. Review have been a balance between finding studies that are similar and appropriate to combine without becoming too focused and summarize the main findings including the strength of evidence for each main outcome; consider their relevance to key groups. Discuss limitations at study and outcome level.

As far as I have seen, there is only one article on MDL. According to one article, it is not very convenient to reach conclusions about the effectiveness of MDL after walking 10min. More article results should be given on this subject. It would be appropriate for the author to correct the inference regarding MDL.

Reviewer #2: The overall goal of this meta-analysis was to evaluate the effect of different taping techniques on pes planus with regards to improving their medial arch deformation. To the reviewer’s point of view, this is an important aspect which has not been comprehensively reviewed before, and it can bring valuable knowledge to the clinicians and researchers. There are a few number of limitations in this manuscript.

Line 14: A verb is required to be added to the sentence “Modified low-dye taping technique could perform better in maintaining navicular height in adults with pes planus after walking for 10 minutes than Augmented low-Dye”

Line 38: As the authors talk about correcting talus position later in the introduction, it is suggested to address the problem that happens to the talus of pes planus, and not just arch deformity. Talus is more a part of rearfoot rather than the medial arch

Line 43: The reviewer suggests using a more appropriate phrase instead of “cushioning ground impact”. It can be probably a phrase like “absorb/ attenuate the impact force”

Line 47-49: “Pes planus is more common in male and adolescents with higher BMI index4, this condition even continues to adulthood without timely and effective intervention”. This sentence is not clear for the reviewer. The authors are requested to re-write it more clearly.

Line 54-56: The authors have suddenly jumped from neuromuscular training to orthoses without explaining anything about orthoses. It is recommended that the authors start with a primary sentence explaining that orthoses are another common treatment for pes planus. However, the orthosis is not efficient in correcting the talus deviation.

Line 67: The authors have mentioned one size for taping. The question is that are they always the same size?

Line 104-106: Although the authors always talk about pes planus as an alternative word to flatfoot and over-pronated foot, they did not use the “pes planus” as a keyword for their searching strategy. Is there a specific reason for that? Is it possible that the authors have missed a study/ studies because of that?

Line 116-120: The inclusion criteria has been mentioned to be navicular drop over 8 mm during walking. Do the authors mean that they excluded the papers which used static foot posture to identify flatfoot (e.g. foot posture index, arch index, footprints, MRI, ...)?

Figure 3: Can the authors make the x-axis range smaller so that the results of each study for 95% of confidence interval can be observed more clearly? For example, the scale of axis for subplot G can be reduced to [-10, 10] instead of [-100, 100].

Line 326-329: I recommend the authors to transfer these sentences to the discussion rather than their results, as it is not the outcome of the current study.

6. PLOS authors have the option to publish the peer review history of their article (what does this mean?). If published, this will include your full peer review and any attached files.

Reviewer #1: No

Reviewer #2: No

---

## [Author Response · Author response to Decision Letter 0]

27 Apr 2021

Dear Editor，

We would like to thank the editor for giving us a chance to resubmit the paper, and also thank the reviewers for giving us constructive suggestions which would help us both in English and in depth to improve the quality of the paper. Here we submit a new version of our manuscript with the title “Revised Manuscript with Track Changes”, which has been modified according to the reviewers’ suggestions. Efforts were also made to correct the mistakes and improve the English of the manuscript. We mark all the changes in red in the revised manuscript.

About the English writing of the manuscript, we ask for native English speaker to revise the paper before it was submitted to the magazine and this time. I don’t know whether it has reached to your magazine’s standard.

Now I answer the questions one-by-one. 

Thank you very much.

Sincerely,

Meihua Tang

Answers to REVIEW 1

1. The authors have made objective clearly stated and clinically relevant and focused study question included.

[Answer]: We thank the reviewer for reading our manuscript carefully and giving the positive comments.

2. Authors has made Comprehensive literature search conducted and searched information sources listed (ie, PubMed, Cochrane database). Terms used for electronic literature search provided, author may add terms ‘Pes Planus’

[Answer]: Thanks to the reviewer on suggesting to properly address the searching terms in our manuscript. In pubmed database, 'pes planus' is included in the 'flat foot' mesh terms, so we didn't add it to the searching words in the initial manuscript, the searching details are attached in the supporting information (S2 Data. Searching terms). However, in other databases like Web of Science and cochrane library, we searched the key word "pes planus" additionally, so we didn't miss the studies potentially. Since there is no explanation of the pubmed mesh word in the original manuscript individually, it is more rigorous to add ‘pes planus’, I have added it in the revised version (Line 116-117).

3. Review included at least two reviewers should search sources for articles relevant to the meta-analysis, and the keywords used in the online searches should be provided in the article.

[Answer]: We agree with the considerate comment. Two independent reviewers searched database including Web of Science, Pubmed, EBSCO, CNKI and Cochrane Library. I have added the searching reviewers in the revised version (Line 112-113). Besides, searching details have presented in the supporting information (S2 data).

4. Inclusion and exclusion criteria have been defined for studies included in the meta-analysis.

[Answer]: We gratefully thanks for the precious time the reviewer spent making constructive remarks.

5. The meta-analysis has provided a table outlining the features of the studies.

[Answer]: We thank the reviewer for reading our paper carefully and giving the positive comments above.

6. Review have been a balance between finding studies that are similar and appropriate to combine without becoming too focused and summarize the main findings including the strength of evidence for each main outcome; consider their relevance to key groups. Discuss limitations at study and outcome level. As far as I see, there is only one article on MDL. According to one article, it is not very convenient to reach conclusions about the effectiveness of MDL after walking 10min. More article results should be given on this subject. It would be appropriate for the author to correct the inference regarding MDL.

[Answer]: Thanks again to the reviewer on suggesting to properly give more articles concerning Modified low-Dye in the early version. As far as I know, there exists only one arcticle that explores the effect of Modified low-Dye maintaining the arch of the foot after walking for 10 minutes, so more experiments are needed in the future to explore the effect after 10 minutes of walking. Meanwhile, I have corrected the conclusion in the revised version. "MLD could possibly performed better than ALD in maintaining immediate navicular height after walking for 10 min, which needs further investigation in the future" （Line 39-40, L514-517)

Answers to REVIEW 2

Reviewer #2: The overall goal of this meta-analysis was to evaluate the effect of different taping techniques on pes planus with regards to improving their medial arch deformation. To the reviewer’s point of view, this is an important aspect which has not been comprehensively reviewed before, and it can bring valuable knowledge to the clinicians and researchers. There are a few limitations in this manuscript.

1. Line 14: A verb is required to be added to the sentence “Modified low-dye taping technique could perform better in maintaining navicular height in adults with pes planus after walking for 10 minutes than Augmented low-Dye”

[Answer]: Considering the reviewer’s valuable suggestion, we have changed to "Modified low-dye taping technique could possibly perform better in maintaining navicular height in adults with pes planus after walking for 10 minutes than Augmented low-Dye, which needs further investigation in the future" in the revised version.(Line 514-517)

2. Line 38: As the authors talk about correcting talus position later in the introduction, it is suggested to address the problem that happens to the talus of pes planus, and not just arch deformity. Talus is more a part of rearfoot rather than the medial arch.

[Answer]: Thank you for the constructive suggestions. As the results showed in the initial manuscript, resting calcaneal stance position after low-Dye taping was closer to a neutral position (Line 327-340), so I have added " Low-Dye could make resting calcaneal stance position closer to neutral position." in the revised version (Line 40-41, L516-517). 

3. Line 43: The reviewer suggests using a more appropriate phrase instead of “cushioning ground impact”. It can be probably a phrase like “absorb/ attenuate the impact force”

[Answer]: We appreciate the considerate comment, we have changed the verb "cushion" to "attenuate" in the revised manuscript (Line 51).

4. Line 47-49: “Pes planus is more common in male and adolescents with higher BMI index, this condition even continues to adulthood without timely and effective intervention”. This sentence is not clear for the reviewer. The authors are requested to re-write it more clearly.

[Answer]: Per the reviewer's request, we have corrected to "Pes planus is more common in adolescent males than females, associated with higher BMI index" in the revised version (Line55-56).

5. Line 54-56: The authors have suddenly jumped from neuromuscular training to orthoses without explaining anything about orthoses. It is recommended that the authors start with a primary sentence explaining that orthoses are another common treatment for pes planus. However, the orthosis is not efficient in correcting the talus deviation.

[Answer]: We thank the reviewer for pointing out this issue. We have changed to " orthoses are another common treatment for pes planus, such as arch supports and taping techniques. However, studies showed that arch supports were not efficient in correcting the talus deviation " in the revised manuscript (Line 62-65).

6. Line 67: The authors have mentioned one size for taping. The question is that are they always the same size?

[Answer]: No, the tapes are not always the same size, as table 1 shows, authors used 2.5 cm, 3.8cm, 5cm, 5.08cm and 7.62cm wide tapes in their studies. It was ambiguous in the initial manuscript, I have changed "2.54–7.62 cm" to "from 2.54 to 7.62 cm" in the revised version (Line 76). 

7. Line 104-106: Although the authors always talk about pes planus as an alternative word to flatfoot and over-pronated foot, they did not use the “pes planus” as a keyword for their searching strategy. Is there a specific reason for that? Is it possible that the authors have missed a study/ studies because of that?

[Answer]: Thanks to the reviewer on suggesting to properly address the searching terms in our manuscript. In pubmed database, 'pes planus' is included in the 'flat foot' mesh terms, so we didn't add it to the searching words in the initial manuscript, the searching details are attached in the supporting information (S2 Data. Searching terms). However, in other databases like Web of Science and cochrane library, we searched the key word "pes planus" additionally, so we didn't miss the studies potentially. Since there is no explanation of the pubmed mesh word in the original manuscript individually, it is more rigorous to add ‘pes planus’, I have added it in the revised version (Line 116-117).

8. Line 116-120: The inclusion criteria have been mentioned to be navicular drop over 8 mm during walking. Do the authors mean that they excluded the papers which used static foot posture to identify flatfoot (e.g. foot posture index, arch index, footprints, MRI, ...)?

[Answer]: We thank the reviewer for this considerate comment, we didn't exclude the papers which used static foot posture to identify flatfoot. The old manuscript mentioned that“such as pronated foot or navicular drop distance over 8 mm or observed a lower medial longitudinal arch height (e.g. excessive pronation) during the stance phase of walking”, which is an example to illustrate one of the three inclusion criteria in 13 articles. In addition, criteria in 4 articles is RCSP> 4°and criteria in another 2 articles is FPI score ≥ 6. I have added the latter two criteria in the revised manuscript(Line 130-133).

9. Figure 3: Can the authors make the x-axis range smaller so that the results of each study for 95% of confidence interval can be observed more clearly? For example, the scale of axis for subplot G can be reduced to [-10, 10] instead of [-100, 100].

[Answer]: Thank you for the constructive suggestions. In the revised manuscript, the scale of axis for Figure 3 subplots A to G have been reduced to [-10, 10]. (Line 258-284)

10. Line 326-329: I recommend the authors to transfer these sentences to the discussion rather than their results, as it is not the outcome of the current study.

Answer: We are appreciative of the reviewer’s suggestion. These sentences have been moved to Line 376-379 in the revised manuscript.

---

## [Decision Letter · Decision Letter 1]

9 Jun 2021

Effects of taping techniques on arch deformation in adults with pes planus: A meta-analysis

PONE-D-20-37311R1

Dear Mr. Hu,

We’re pleased to inform you that your manuscript has been judged scientifically suitable for publication and will be formally accepted for publication once it meets all outstanding technical requirements.

Kind regards,

Slavko Rogan

Academic Editor

PLOS ONE

Additional Editor Comments (optional):

Reviewers' comments:

Reviewer's Responses to Questions

**Comments to the Author**

1. If the authors have adequately addressed your comments raised in a previous round of review and you feel that this manuscript is now acceptable for publication, you may indicate that here to bypass the “Comments to the Author” section, enter your conflict of interest statement in the “Confidential to Editor” section, and submit your "Accept" recommendation.

Reviewer #2: All comments have been addressed

2. Is the manuscript technically sound, and do the data support the conclusions?

Reviewer #2: Yes

3. Has the statistical analysis been performed appropriately and rigorously? 

Reviewer #2: Yes

4. Have the authors made all data underlying the findings in their manuscript fully available?

Reviewer #2: Yes

5. Is the manuscript presented in an intelligible fashion and written in standard English?

Reviewer #2: Yes

6. Review Comments to the Author

Reviewer #2: My comments have been adequately addressed. The authors could modify the manuscript based on the comments and suggestions of the reviewers. I think this meta-analysis can bring valuable knowledge to the

clinicians and researchers.

7. PLOS authors have the option to publish the peer review history of their article (what does this mean?). If published, this will include your full peer review and any attached files.

Reviewer #2: No

---

## [Editor Report · Acceptance letter]

17 Jun 2021

PONE-D-20-37311R1 

Effects of taping techniques on arch deformation in adults with pes planus: A meta-analysis 

Dear Dr. Hu:

I'm pleased to inform you that your manuscript has been deemed suitable for publication in PLOS ONE. Congratulations! Your manuscript is now with our production department. 

Kind regards, 

on behalf of

Dr. Slavko Rogan 

Academic Editor

PLOS ONE